# Integrated Metabolomic and Transcriptomic Analysis Reveals That Amino Acid Biosynthesis May Determine Differences in Cold-Tolerant and Cold-Sensitive Tea Cultivars

**DOI:** 10.3390/ijms24031907

**Published:** 2023-01-18

**Authors:** Yaohua Cheng, Qiuyan Ban, Junlin Mao, Mengling Lin, Xiangxiang Zhu, Yuhui Xia, Xiaojie Cao, Xianchen Zhang, Yeyun Li

**Affiliations:** 1State Key Laboratory of Tea Plant Biology and Utilization, School of Tea & Food Science, Anhui Agricultural University, Hefei 230036, China; 2School of Horticulture, Henan Agriculture University, Zhengzhou 450002, China

**Keywords:** *Camellia sinensis*, cold acclimation, transcriptomic, metabolomic, amino acid biosynthesis, arginine

## Abstract

Cold stress is one of the major abiotic stresses limiting tea production. The planting of cold-resistant tea cultivars is one of the most effective measures to prevent chilling injury. However, the differences in cold resistance between tea cultivars remain unclear. In the present study, we perform a transcriptomic and metabolomic profiling of *Camellia sinensis* var. “*Shuchazao*” (cold-tolerant, SCZ) and *C*. *sinensis* var. *assamica* “*Yinghong 9*” (cold-sensitive, YH9) during cold acclimation and analyze the correlation between gene expression and metabolite biosynthesis. Our results show that there were 51 differentially accumulated metabolites only up-regulated in SCZ in cold–acclimation (CA) and de–acclimation (DA) stages, of which amino acids accounted for 18%. The accumulation of L-arginine and lysine in SCZ in the CA stage was higher than that in YH9. A comparative transcriptomic analysis showed an enrichment of the amino acid biosynthesis pathway in SCZ in the CA stage, especially “arginine biosynthesis” pathways. In combining transcriptomic and metabolomic analyses, it was found that genes and metabolites associated with amino acid biosynthesis were significantly enriched in the CA stage of SCZ compared to CA stage of YH9. Under cold stress, arginine may improve the cold resistance of tea plants by activating the polyamine synthesis pathway and CBF (C-repeat-binding factor)–COR (cold-regulated genes) regulation pathway. Our results show that amino acid biosynthesis may play a positive regulatory role in the cold resistance of tea plants and assist in understanding the cold resistance mechanism differences among tea varieties.

## 1. Introduction

The tea plant is a perennial economic woody crop. Its leaf products are used to make popular nonalcoholic beverages worldwide [1]. Tea plants are exposed to a wide variety of environmental stresses, such as drought [2], heavy metal toxicity [3], nutritional deficiency [4], and low temperatures [5]. Low temperatures (0 °C < *T*_mean_ ≤ 4 °C) represent a common and major natural stress factor in most tea cultivation areas in China, seriously affecting the yield and quality of tea [6]. At low temperatures, the photosynthesis of tea plants is damaged [7] and the aromatic components of the leaves also decrease significantly [8]. Understanding the complexity of cold resistance in tea plants and improving their resistance levels as a result is, therefore, of great importance.

Advances in functional genomics have led to the elucidation of the molecular mechanisms underlying the low-temperature responses of tea plants. Studies have shown that the ethylene and calcium signaling pathways mediated by mitogen-activated protein kinase (MAPK) are followed by the induction of the inducer of CBF expression (ICE)–C–repeat binding factor (CBF)–cold–responsive (COR) signaling pathway to enhance the cold tolerance of tea plants [5]. In addition, many genes related to sugar metabolism, antioxidant defense, plant hormone signaling, transcription factors, and amino acid synthesis that respond to cold stress in tea plants have been identified [9,10,11]. Previous studies showed that cold treatment significantly induced *CsCBFs* genes in tea plants, and the overexpression of *CsCBF3* and *CsCBF5* enhanced the cold tolerance of *Arabidopsis* compared with that of a wild-type *Arabidopsis* species [12,13]. Some newly identified genes, such as *CsCHXB* and *CsUGT78A15,* have been explicitly associated with a functional role in cold tolerance [14,15].

Metabolites are directly involved in the cold tolerance of tea plants. For instance, carbohydrates and amino acids accumulate under low temperatures during the plant stress response and act as osmotic protection agents. Under natural overwintering conditions, raffinose, maltose, glucose, and fructose were accumulated in tea leaves [16]. A metabolomic analysis showed a significant accumulation of glucose 6-phosphate, fructose 1-phosphate, and anthocyanin monomers in *Shuchazao* under cold stress [17]. For survival under low temperatures, more energy carriers have to generate lipids, amino acids, membrane components, and other molecules to improve cell membrane fluidity and structural rearrangement [18]. Proline and free amino acids levels have been positively correlated with cold resistance in tea plants [19]. The application of exogenous γ-aminobutyric acid either directly or indirectly stimulated flux into amino acid and caffeine biosynthesis and regulated the plant energy budget and improve the cold resistance of tea plants [20]. Similar changes have also been observed in wheat [21] and tobacco [22]. Studies have shown that higher levels of flavonoids, such as kaempferol, quercetin, and anthocyanins, are formed in tea plants under cold stress to resist oxidative damage [23].

Cultivated varieties of tea plants mainly include *Camellia sinensis* var. *sinensis* (CSS) and *Camellia sinensis* var. *assamica* (CSA) in China [24]. CSS is a shrub-type tea plant that can withstand low temperatures of −12~−16 °C and is mainly distributed in the cold climate regions of the tea-growing areas. CSA is an arbor-type tea plant that is severely damaged at around −6 °C and is distributed primarily in warm tea gardens. Although there have been many reports on the mechanisms of cold resistance in tea plants, almost all the varieties studied thus far belong to CSS. In addition, the cold resistance mechanism differences among tea cultivars are still unclear. In this study, we report that the biosynthesis of amino acids may determine differences in the cold resistance of tea plants as revealed by a comparative metabolomic and transcriptomic analysis between the cold-tolerant tea plant variety SCZ and the cold-sensitive variety YH9. We additionally characterize the role of arginine as being important factor determinant of cold tolerance in tea plants.

## 2. Results

### 2.1. Differences in Cold Resistance between SCZ and YH9

The experimental temperature treatment procedure is shown in Figure 1A, where SCZ and YH9 tissue culture seedlings were used for cold acclimation. We then compared the freezing resistance of cold-acclimated leaves from different cultivars via a relative electrical conductivity measurement (REC). In the non-acclimation (NA) group, the REC of the two varieties was about 50%, indicating that freezing caused serious cell damage. However, after cold acclimation (CA) and de-acclimation (DA) treatments, the REC of SCZ was significantly lower than that of YH9, indicating that the frost resistance of SCZ had greatly improved. After CA treatments, the chlorophyll fluorescence (Fv/Fm) value was higher in SCZ than in YH9 (Appendix A), and it increased by 68.67% under the −4 °C treatment, which was significantly higher than that of YH9 (Figure 1C). These results suggest that cold acclimation can improve the cold resistance of two tea cultivars to different degrees, and SCZ is more resistant to cold than YH9.

### 2.2. Differential Metabolites between SCZ and YH9 during Cold Acclimation

The metabolites of SCZ and YH9 during cold acclimation were detected, and the differences in the metabolites between the cultivars or conditions were analyzed. A metabolomics analysis showed that a total of 359 differential metabolites were identified, and all the metabolites were divided into groups, namely lipids and lipid-like molecules (115); flavonoids (64); benzenoids (45); organoheterocyclic compounds (36); amino acids (18); organic acids and derivatives (12); nucleosides, nucleotides, and analogues (12); saccharides (4); and others (53) (Appendix A). A PCA analysis of metabolites from control and cold-acclimated groups of SCZ and YH9 revealed that six biological replicates were similar at each time point and that PC1 could completely distinguish between the two cultivars, indicating differences in metabolic levels between SCZ and YH9 (Appendix A). The significant separation of PC2 in the NA and CA stages indicated metabolic changes under cold acclimation (Figure 2A). Based on the quantitative analyses of all the detected metabolites and the fold-change threshold, a total of 131 and 120 differentially accumulated metabolites were obtained for comparing SCZ_NA vs. YH9_NA and SCZ_CA vs. YH9_CA, respectively (Figure 2B). Among all the metabolites detected, 43 flavonoids were significantly different in SCZ_NA vs. YH9_NA, and 37 flavonoids were significantly different in SCZ_CA vs. YH9_CA, indicating that flavonoids were the main substances that were different between SCZ and YH9 (Appendix A). For the two comparisons of SCZ_NA vs. YH9_NA and SCZ_CA vs. YH9_CA, 44 differentially accumulated metabolites were up-regulated in SCZ_CA vs. YH9_CA and YH9_NA and SCZ_CA vs. YH9_CA (Appendix A), including lysine and L-arginine (Appendix A). These differentially accumulated metabolites may be related to the difference in cold resistance between SCZ and YH9.

A total of 61 and 55 differentially accumulated metabolites were identified in SCZ and YH9 in the CA stage, respectively. Compared with NA, 81 and 41 metabolites were differentially accumulated in the leaves of SCZ and YH9 in the DA stage (Figure 2A). In Figure 2C, we summarize the differences in metabolites between the different comparison groups (using Venn diagrams). A total of 51 differential accumulated metabolites were specifically up-regulated in SCZ at CA and DA stages, mainly amino acids (18%); flavonoids (18%); lipids and lipid-like molecules (14%); benzenoids (14%); and organoheterocyclic compounds (10%). A cluster analysis of metabolites showed significant differences between SCZ and YH9 in the CA and DA stages (at least one stage) using the mean area of repetitive biological peaks based on the Euclidean distance (Figure 2D). The amino acid differentially accumulated metabolites were most enriched in SCZ. The differential accumulation of L-histidine, L-arginine, lysine, glutamine, leucine, and tyrosine was the highest in the CA stage of SCZ, and the specific accumulation of L-5-oxoproline, L-histidine, L-phenylalanine, lysine, glutamine, and leucine was highest in the DA stage. These results show that cold acclimation could induce a higher accumulation of amino acids in cold-resistant SCZ. The metabolites were significantly up-regulated in the CA stage of SCZ and YH9, mainly saccharides (sucrose, trehalose, and D-(+)-raffinose) and lipids and lipid-like molecules (alpha-linolenic acid, gamma-linolenic acid, and linoleic acid), indicating that these differential metabolites were involved in the cold acclimation of the tea plants. Only a small number of flavonoids accumulated differentially during cold acclimation, among which the accumulation of cyanidin-3-glucoside and astragalin was involved in YH9 during cold acclimation, while vitexin-2″-o-rhamnoside was involved in SCZ in the CA stage. These results show that although flavonoids varied significantly among varieties, only a small number of flavonoids were induced during cold acclimation. In addition, some metabolites showed opposite differential accumulation patterns in the CA stages of SCZ and YH9; for example, L-glutathione, which was significantly up-regulated in SCZ but significantly down-regulated in YH9.

### 2.3. Transcriptomic Analysis of SCZ and YH9 during Cold Acclimation

A total of 7.17 billion high-quality clean reads were obtained from the RNA-seq data set for 18 samples. All the clean reads were mapped to the tea reference genome (http://pcsb.ahau.edu.cn:8080/CSS/ (accessed on 1 June 2017)), ranging from 97.10% to 97.78%. The clean reads shared a GC content of approximately 45% (Appendix A). According to the PCA, the first two principal components (PC1 and PC2) could completely to distinguish our combinations of cultivars and treatment (Appendix A). The PCA showed that the CA stage of separation could be observed along PC1, and differences between cultivars that had obvious separation could be observed along PC2. In addition, the three biological replicates were projected closely in the ordination space, which suggested a satisfactory correlation among the replicates (Figure 3A).

The six comparisons (SCZ _ CA vs. SCZ _ NA, YH9 _ CA vs. YH9 _ NA, SCZ _ DA vs. SCZ _ NA, YH9 _ DA vs. YH9 _ NA, SCZ _ NA vs. YH9 _ NA and SCZ _ CA vs. YH9 _ CA) had 6812, 8798, 728, 1044, 4833, and 5749 DEGs, respectively: 2963, 3997, 232, 527, 2519, and 2556 up-regulated, and 3849, 4801, 496, 517, 2314, and 3193 down-regulated (Figure 3B). These results show that the gene expression levels in cold-tolerant and cold-sensitive tea plants changed significantly during cold acclimation. A Venn diagram analysis showed that 18 common unigenes were up-regulated in SCZ and YH9, while 87 common unigenes were down-regulated (Figure 3C). In addition, 1220 up-regulated DEGs were found uniquely in SCZ during cold acclimation, while 2443 unique up-regulated DEGs were obtained in YH9. These up-regulated DEGs unique to SCZ might be related to cold tolerance, while the up-regulated DEGs unique to YH9 might be correlated with cold sensitivity. For the two comparisons SCZ _ NA vs. YH9 _ NA and SCZ _ CA vs. YH9 _ CA, there were 3119 DEGs found only in SCZ _ CA vs. YH9 _ CA, which might be involved in the response of the tea varieties to the CA stage. To validate the RNA-seq data, changes in the gene expression of nine randomly selected DEGs were analyzed via real-time quantitative RT-PCR (RT-qPCR). Although the fold changes of selected genes differed between RNA-seq and RT-qPCR, a similar trend was observed in both analyses (Appendix A).

To further determine the specific expression pathways of the two tea varieties, a functional enrichment analysis of the genomes was performed using KEGG (Figure 3D). The KEGG pathway enrichment analysis of the DEGs showed that there were 12 common pathways between the comparisons of SCZ_CA vs. NA and YH9_CA vs. NA, among which “glycosphingolipid biosynthesis—ganglio series” and “other types of o-glycan biosynthesis” were the most significantly enriched pathways, which indicated that these pathways might be the most relevant to the cold acclimation of tea plants. Compared with the set obtained for YH9_CA vs. NA, four significantly enriched pathways were uniquely observed in the set for SCZ_CA vs. NA, which were “fatty acid elongation”, “starch and sucrose metabolism”, “arginine biosynthesis”, and “butanoate metabolism”. Amino acid biosynthesis was specifically enriched during the cold acclimation of SCZ, which was consistent with the metabolomics analysis results. In addition, “glycine, serine and threonine metabolism” were enriched by comparing SCZ _NA vs. YH9 _NA and SCZ _CA vs. YH9 _CA, “arginine and proline metabolism” and “D-alanine metabolism” were specifically enriched in SCZ _CA vs. YH9 _CA, and SCZ _NA vs. YH9 _NA was specifically enriched in “tyrosine metabolism”, “arginine biosynthesis”, and “alanine, aspartate, and glutamate metabolism”. All the above-mentioned pathways might be correlated with the cold resistance of tea plants.

### 2.4. Integrated Analysis of Transcriptomic and Metabolomic of SCZ and YH9

To further examine the relationship between DEGs and the differential accumulation metabolites in tea plants responsive to cold acclimation, a co-expression network analysis of the transcriptomic and metabolomic data was conducted to compare SCZ_CA and YH9_CA. Screening of DEGs and differentially accumulated metabolites by Pearson correlation coefficients of >0.8 and a heatmap analysis revealed 97 metabolites regulated by 4571 genes in the CA stage, and Figure 4A vividly displays their clustering characteristics. A combined KEGG analysis of DEGs and differential accumulation metabolites was also conducted. Results showed that the DEGs and differential accumulation metabolites enriched in the pathways of “purine metabolism”, “biosynthesis of amino acids”, and “tyrosine metabolism” exhibited consistent expression patterns with significant differences (*p*−value < 0.05). Among them, only the “amino acid biosynthesis” pathway was significantly different (*p*−value < 0.01), further confirming the conclusion that SCZ was mainly positively regulated by amino acid biosynthesis in the CA stage (Figure 4B). Therefore, we detected the amino acid contents of the two tea plant varieties in the CA stage (Appendix A). In the CA stage, the aspartic acid, ornithine, and arginine contents of SCZ were significantly higher than those of YH9. These results were consistent with the accumulation of arginine in the metabolomics. Figure 4C shows the construction of the gene–metabolite association network for the amino acid biosynthesis pathway. A total of 21 unigenes involved in amino acid biosynthesis and metabolism were highly positively or negatively correlated with lysine and L−arginine, including primary−amine oxidase (*AOC*), 2,3−bisphosphoglycerate−independent phosphoglycerate mutase (*gpmI*), S−adenosylmethionine synthetase (*MAT*), cyclohexadienyl dehydratase (*pheC*), spermine synthase (*SPMS*), and ketol−acid reductoisomerase (*ilvC*) genes.

### 2.5. Expression of Genes and Changes of Metabolites in Amino Acid Biosynthesis in the CA Stage

The biosynthesis pathways of amino acids in tea plants in response to cold acclimation are illustrated in detail by combining the KEGG databases (Figure 5). Amino acid biosynthesis originates from the glycolysis pathway. *PFK*, *PGK*, *ENO, PK,* and *CS* are key genes in the glycolysis/gluconeogenesis pathway, catalyzing the formation of glyceraldehyde 3-phosphate, 3-phospho-D-glycerate, phosphoenolpyruvate, pyruvate, and citrate. Our data show that SCZ_CA has higher *HK*, *PGAM*, *PK*, and *CS* gene expression levels than YH9_CA, which may lead to the continuous accumulation of catalytic reaction products. However, the direct products of these catalytic reactions were not detected in this study.

The amino acid biosynthesis pathway is divided into five branches: histidine, aromatic amino acid (phenylalanine and tyrosine), branched-chain amino acid (leucine), and basic amino acid (lysine and arginine) biosynthesis pathways. The transcriptomic analysis screened the critical genes involved in amino acid synthesis and degradation pathways in tea plants. In comparing SCZ_CA and YH9_CA, high expression levels of the *RPE*, *rpiA*, *PRPS*, *hisG,* and *hisC* genes were observed in the cold-tolerant SCZ, which were consistent with the accumulation of histidine, the catalytic reaction product. In the aromatic amino acid biosynthesis pathway, we observed that the *aroF*, *aroE*, *aroK*, *aroC*, *tyrA2,* and *pheC* genes were highly expressed in SCZ, which was consistent with the accumulation of phenylalanine and tyrosine. In the biosynthesis pathway of aspartate−derived amino acids, the expression levels of genes encoding *GOT1*, *AK,* and *dapB* increased, which was consistent with the significant accumulation of lysine. In the pathway of glutamine and arginine biosynthesis, we observed that the expression levels of genes encoding *GLT1*, *argC*, *argD*, *argE*, *OTC*, and *ASS1* in SCZ_CA were higher than those in YH9_CA. By contrast, the genes encoding *argB* and *GLUL* showed opposite expression patterns. The expression of these genes may be related to arginine and glutamine accumulation.

### 2.6. Exogenous Application of Arginine Improved Cold Tolerance in Tea Plants

Through the comprehensive transcriptomic and metabolomic of a cold-tolerant variety of SCZ and a cold-sensitive variety of YH9 under cold acclimation, it was found that the synthesis of amino acids was the main factor for the difference in cold resistance of tea plants, among which arginine had the most significant difference. To validate the role of arginine in the cold stress of the tea plants, assays were performed under an exogenous supply of arginine (1% *w*/*v*). Arginine application significantly alleviated the cold stress-induced damage of SCZ (Figure 6A). Consistently, the peroxidase (POD) activities of tea leaves sprayed with arginine at −4 °C increased by 37.50% (Figure 6B). Catalase (CAT) activities significantly increased by one-fold.

UPLC measured the polyamine content in tea leaves after spraying the foliage with arginine. The concentrations of putrescine, spermidine, and spermine were significantly increased. The levels of spermidine and spermine were 16.82% and 93.60% higher than they were under cold stress, respectively, and the putrescine content increased by 7.63-fold (Figure 6C). The expression of genes related to the polyamine synthesis pathway was analyzed. The results showed that expression levels of arginine decarboxylase (*CsADC*), agmatine iminohydrolase (*CsAIH*), N-carbamoylputrescine amidohydrolase (*CsCPD*), spermidine synthase (*CsSPDS*), and spermine synthase (*CsSPMS*) were significantly increased following treatment with arginine after cold stress and were significantly higher than in tea leaves exposed to cold stress alone, especially the expression levels of *CsADC* (Figure 6D).

The CBF–COR regulation pathway is an important cold regulation pathway in tea plants. Our results showed that spraying arginine had a regulatory effect on this pathway. The exogenous application of arginine enhanced the expression of *CBFs* under cold stress (Figure 6E). The expression levels of *CsCBF2* and *CsCBF3* were significantly increased by 3.46- and 7.52-fold, while *CsCBF1* expression was most significantly increased by 28.94-fold. The transcription of three downstream COR genes (*CsLEA*, *CsLEA3L-1*, and *CsCIP*) increased significantly following the exogenous application of arginine.

## 3. Discussion

Cold stress is an uncontrollable climate factor that can impair the growth of tea plants, thus limiting tea yield and quality [25]. The modes of physiological and molecular responses to cold stress have been demonstrated in tea plants [26]. However, these studies have mainly been conducted on a single tea variety, and the investigation of the cold resistance of a single tea variety may have specific limitations. In this study, we compared the transcriptional and metabolic profiles of SCZ (*Camellia sinensis* var. ”*Shuchazao*”) and YH9 (*C. sinensis* var. *assamica* “*Yinghong 9*”) during cold acclimation and found that amino acid biosynthesis may be the pathway responsible for the differences in the cold resistance of tea cultivars.

At the transcriptional level, cold-sensitive tea cultivars were more disturbed by cold acclimation [27], since more CA-responsive genes were identified in YH9 (Figure 3B). However, at the metabolic level, we found more cold acclimation-responsive metabolites in the cold-tolerant SCZ (Figure 2B), suggesting that post-transcriptional regulation plays a vital role in the cold tolerance of tea plants. Free amino acids play an essential role in plant development and metabolism [28]. Amino acids are nitrogenous organic compounds, the main products of nitrogen and sulfur metabolism under environmental stress [29]. The accumulation of amino acids is frequently observed in many plant species subjected to cold stress [30,31,32]. A study found the amino acid accumulation of cold-tolerant tea varieties is significantly higher than that of cold-sensitive tea varieties under cold acclimation [33]. The accumulation of amino acids and amino acid derivatives contributes to the low temperature acclimation of *Arabidopsis* and *Thellungiella*, including alanine, asparagine, β-alanine, histidine, isoleucine, phenylalanine, serine, threonine, and valine [34]. In this study, we observed the specific accumulation of L-5-oxoproline, L-histidine, L-phenylalanine, L-arginine, lysine, glutamine, leucine, and tyrosine in the cold-tolerant SCZ during cold acclimation (at least one stage) (Figure 2D). We attended the enrichment of pathways related to amino acid biosynthesis in SCZ, for example, the arginine biosynthesis pathway (Figure 3D). These results indicated that the accumulation of amino acids during cold acclimation mainly occurred in cold-tolerant SCZ.

Lysine is considered the most important essential amino acid, affecting the utilization of other amino acids. Lysine plays a role in the abiotic stress response mainly through glycosamine pathway catabolism [35]. It induces the jasmonic acid signaling pathway and tryptophan metabolism in the stress response [36]. In our study, SCZ showed a significant increase in lysine content in the CA stage, which was 5.61-fold higher than YH9. Aspartate aminotransferase (*GOT1*) and aspartate kinase (AK) are essential enzymes that catalyze aspartate biosynthesis [37,38]. Studies have shown that the average activities of aspartic acid aminotransferase (*GOT1*) and aspartic acid kinase (AK) in rice grains were significantly increased under abiotic stress, and the lysine content was also increased considerably [39]. 4-Hydroxy-tetrahydrodipicolinate reductase (dapB) is the core enzyme of lysine biosynthesis, which can synthesize lysine in bacteria and plants, but its expression in stress is unknown [40]. In the CA stage, the expression levels of the *GOT1*, *AK,* and *dapB* genes involved in the lysine biosynthesis pathway were higher in SCZ than in YH9, indicating that the lysine biosynthesis pathway played an important role in improving the cold tolerance of the tea plant (Figure 5). Arginine is a metabolically versatile amino acid with a high N/C ratio and plays a vital role in stress defense mechanisms [41]. For example, a high level of arginine accumulation was observed in wild watermelon (*Citrullus lanatus*) under drought stress [42]. After cold treatment, α-ketoglutarate derivatives (arginine) accumulated in strawberry leaves [43]. The data showed a significant accumulation of arginine in SCZ in the CA stage, and the arginine content in SCZ was 11.92-fold that of YH9, which meant arginine showed the most variation of all the amino acid substances among the tea cultivars. Similar findings were also recorded in peanut varieties (*Zhonghuahei1* and *Zhongkaihua151*) [44]. Arginine is formed by ornithine carbamoyltransferase (*OTC*) and synthesized by argininesuccinate synthase (*ASS1*) [41]. After arsenic stress, a different expression of *OTC* protein in maize leaves was inoculated with arbuscular mycorrhizal fungi (AMF) [45]. The plant argininesuccinate synthase gene *Gh ASS1* was cloned from cotton, suggesting its possible role in regulating plant stress resistance [46]. In the CA stage, the expression of the *OTC* gene involved in arginine biosynthesis in SCZ was higher than in YH9. By contrast, the difference in *ASS1* genes between SCZ and YH9 was not noticeable, indicating that the *OTC* gene may play an essential role in arginine accumulation (Figure 5).

Glutathione is an antioxidant commonly found in plants and plays a crucial role in maintaining tissue antioxidant defenses and regulating redox-sensitive signal transduction [47]. Studies have shown that as a signaling molecule, glutathione has cross-talk with other signaling molecules to enhance plant adaptation to adversity [48]. In this study, glutathione accumulation was significantly higher in SCZ_CA than in YH9_CA. The synthesis of glutathione occurs via two enzymatic steps; The first step in the synthesis is catalyzed by γ-glutamylcysteine synthetase (GCLC), and the second is catalyzed by glutathione synthetase (*GS*) [49]. It has been reported that *GCLC* and *GS* transcription levels in *Medicago falcata* increased under cold treatment [50]. In addition, the chilling tolerance of tomato plants after *GCLC* and *GS* co-silencing decreased. In this study, the expression of the *GCLC* and *GS* genes involved in glutathione biosynthesis in SCZ was higher than in YH9 (Figure 5).

In general, cold-tolerant varieties/lines possess higher levels of endogenous PAs relative to cold-sensitive species in response to cold stress [51]. Under low-temperature conditions, an apple’s putrescine and spermidine content is higher than that under normal conditions [52]. In potatoes, the overexpression of *ADC1* lines increases cold tolerance by activating the CBF regulatory pathway and there may be an interaction between CBFs and putrescine [53]. It was also shown that overexpression of the *CsICE1* gene promotes polyamine accumulation through interaction with *ADC* genes and plays a positive role in cold tolerance [54]. Arginine is the precursor of polyamine biosynthesis mediated via arginine decarboxylase (*ADC*) [53]. Studies reported that ADC-catalyzed putrescine synthesis contributes to potato freezing tolerance [55]. This study showed that exogenous arginine supply increased the expression of the *CsADC* and *CsSPMS* genes, and the putrescine content in tea plants under cold stress, thereby enhancing the cold stress tolerance of the tea plants (Figure 6). The CBF–COR regulatory pathway represents a critical pathway in the cold signaling mechanisms of many species [27,55]. In this study, three *CBF* genes were screened via a transcriptomic analysis and all the *CBF* genes members were induced under cold stress following treatment with arginine. The expression levels of *CsCBF1* were dramatically increased by approximately 30-fold. The expression of *CBF* genes was positively correlated with cold tolerance, and the expression of downstream CORs was activated by binding with DRE/CRT cis-acting elements [56]. In our study, three putative CORs regulated by *CBF* genes were also up-regulated. In *Arabidopsis*, studies involving CBF regulators reported that the levels of fructose, glucose, GABA, putrescine, and raffinose were regulated by CBFs [57]. In addition, arginine may also increase NO levels [58], which in turn act as a signal to activate *MPK1*/*2* transcripts and enhance the expression of *CBF1*, thereby improving the cold resistance of tomatoes [59]. However, the transgenic system of tea plant has not been established yet, so the number of functionally characterized genes is still small. In this study, the key genes of amino acid metabolism pathway in tea leaves are almost not cloned and identified. Therefore, the annotation and pathway of these genes can only be predicted based on other related species.

## 4. Materials and Methods

### 4.1. Plant Materials and Treatments

Tissue culture seedlings of the cold-tolerant *Camellia sinensis* var. “*Shuchazao*” and the cold-sensitive *C. sinensis* var. *assamica cv.* “*Yinghong 9*” were obtained from the Anhui Agricultural University, Hefei, China. Plants propagated from cuttings were transferred to a growth chamber under temperature cycles of 25 °C during the day and 20 °C at night, 12 h photoperiod, and 70% relative humidity for 10 months. Subsequently, seedlings with a height of 4–5 cm were subjected to cold acclimation treatments (Figure 1A).

Tissue culture seedlings of tea plants were collected from the aforementioned growth chamber and used as non-acclimation controls (NA). SCZ and YH9 plants were exposed to low temperatures (10 °C daytime, 4 °C nighttime) for 7 days during chilling acclimation. Subsequently, freezing acclimation was conducted by exposing SCZ and YH9 plants to lower temperatures (4 °C daytime, 0 °C nighttime) for another 7 days (CA). Lastly, the plants were exposed to normal temperatures (25 °C daytime, 20 °C nighttime) for 7 days for de-acclimation (DA). After each treatment, two to three leaves of the tea seedlings were collected afresh and immediately frozen in liquid nitrogen at −80 °C until use. Each tea cultivar was subjected to two independent experiments, and transcriptional high-throughput sequencing and metabolic group detection were performed at Hangzhou Lianchuan Biotechnology Co., Ltd. (Hangzhou, China).

### 4.2. Metabolite Identification and Quantification

Metabolites were extracted with 50% methanol buffer. Briefly, 20 µL of the sample was extracted with 120 µL of precooled 50% methanol, vortexed for 1 min, and incubated at room temperature for 10 min. The extract was then stored overnight at −20 °C. After centrifugation at 4000× *g* for 20 min, the supernatants were transferred into new 96-well plates. In addition, pooled QC samples were also prepared by combining 10 μL of each extraction mixture. The samples were stored at −80 °C before LC–MS analysis. The metabolite profiling experiments were repeated to have six biological replicates. All samples were acquired by the LC–MS system depending on machine parameters.

Chromatographic separations were performed via ultra-performance liquid chromatography (SCIEX, UK). An ACQUITY UPLC T3 column (100 mm × 2.1 mm, 1.8 µm, Waters, UK) was used for reversed-phase separation. The column oven was maintained at 35 °C. The flow rate was maintained at 0.4 mL/min. The mobile phase consisted of a mixture of solvent A (water, 0.1% formic acid) and solvent B (acetonitrile, 0.1% formic acid). The linear gradient program for elution was as follows: 0–0.5 min, 5% B; 0.5–7 min, 5% to 100% B; 7–8 min, 100% B; 8–8.1 min, 100% to 5% B; 8.1–10 min, 5% B. The injection volume for each sample was 4 µL.

The metabolites eluted from the column were detected using a TripleTOF5600plus (SCIEX, UK) high-resolution tandem mass spectrometer. The Q-TOF was operated in both positive and negative ion modes. The curtain gas was set at 30 psi, the ion source gas 1 was set to 60 psi, and the ion source gas 2 was set to 60 psi. The interface heater temperature was maintained at 650 °C. The ion Spray Voltage Floating of positive and negative ion modes was set at 5000 V and −4500 V, respectively. The mass spectrometry data was acquired in information-dependent acquisition (IDA) mode. TOF mass ranged from 60 to 1200 Da. The survey scans were acquired in 150 ms, and as many as 12 product ion scans were collected if exceeding a threshold of 100 counts per second (counts/s) with a 1+ charge state. The total cycle time was fixed to 0.56 s. Four bins were summed for each scan at a pulsar frequency of 11 kHz by monitoring the 40 GHz multichannel TDC detector via four-anode/channel detection. Dynamic exclusion was set for 4 s. During the acquisition, mass accuracy was calibrated for every 20 samples. Furthermore, to evaluate the LC-MS stability during the whole acquisition, a quality control sample pool was acquired after every 10 samples.

The acquired MS data pretreatments, including peak picking, peak grouping, retention time correction, second peak grouping, and annotation of isotopes and adducts, were performed using XCMS software (http://metlin.scripps.edu/download/ (accessed on 1 June 2017)). The online KEGG (http://www.kegg.jp/ (accessed on 1 June 2017)) and HMDB (http://www.hmdb.ca/ (accessed on 1 June 2017)) databases were used to annotate the metabolites by matching the exact molecular mass data (m/z) of samples with those from databases. If the mass difference between observed and database value was less than 10 ppm, the metabolite was annotated, and the molecular formula of metabolites was further identified and validated by the isotopic distribution measurements. Since there are many isomeric metabolites in the database, the first-level identification results often have many cases of one m/z corresponding to multiple metabolites. In this analysis, the metabolite secondary mass spectrometry library from in-house was used to match the metabolite secondary mass spectrometry data of the samples to obtain metabolite identification results with higher confidence. Student *t*-tests were conducted to detect differences in metabolite concentrations between two phenotypes. The *p* value was adjusted for multiple tests using the FDR (False discovery rate). Supervised PLS-DA was conducted through metaX to discriminate the different variables between groups. The VIP (variable important for the projection) value was calculated. A VIP cut-off value of 1.0 was used to select important features. The differentially expressed metabolites simultaneously satisfy, (1) fold-change ≥ 2 or fold-change ≤ 0.5; (2) q-value ≤ 0.05; and 3) VIP ≥ 1.

### 4.3. RNA Extraction and Sequencing

The RNA of the total samples was isolated and purified using TRIzol Reagent Kit (Invitrogen, Carlsbad, CA, USA). The amount and purity of the total RNA was then quality controlled using NanoDrop ND-1000 (NanoDrop, Wilmington, DE, USA). The integrity of the RNA was checked by Agilent 2100, and a RIN number > 7.0 was considered as the passing standard.

The quality of the total RNA was inspected followed by enrichment of the eukaryotic mRNA using magnetic beads connected with oligo (dT). The extracted mRNA was randomly cut into short fragments using fragmentation buffer. The fragmented mRNA was used as a template to synthesize a strand of cDNA with random hexamers and was treated with buffer, dNTPs, RNaseH, and DNA polymerase I to perform a two-strand cDNA synthesis. AMPure XP beads were used to purify double-stranded products. T4 DNA polymerase and Klenow DNA polymerase activities were measured to determine the repair of the sticky ends of DNA to blunt ends, followed by addition of base A to the 3′end. A linker was then added followed by fragment selection using AMPureXP beads and PCR amplification to enrich the final sequencing library. Transcriptome sequencing was performed on an Illumina Hiseq4000 platform with a read length of 150 bp. The reads are publicly available at the GEO database of NCBI under the project accession number GSE216311.

### 4.4. Transcriptome Analysis

Raw reads generated from the Illumina Hiseq 4000 platform were preprocessed to clean adapter sequences. Reads containing > 5% unknown bases were filtered out and low-quality reads (>20% with a base mass fraction of 10) were removed. Transcript assembly was constructed using the publicly available tea tree genome as a reference sequence [60]. Then, the clean reads were mapped to the genome using Hisat2 software and assembled using StringTie1.3 with default parameters. Gene expression levels were estimated as fragments per kilobase per transcript per million mapped reads (FPKM). Pearson’s correlations of samples were performed on variance-stabilized transformed values, and principal component analysis (PCA) was performed using the OmicStudio tools at https://www.omicstudio.cn/tool (accessed on 1 June 2017). Differentially expressed genes were defined as exhibiting at least two-fold change in transcript abundance and a FDR (false discovery rate) of <0.05.

### 4.5. Quantitative Determination of Amino Acids

The tea samples were weighed 0.05 g in a 5 mL centrifuge tube, mixed with 2 mL of 4% sulfosalicylic acid, centrifuged (12,000 r/min, 30 min) after 40 min of ultrasonic extraction, and the supernatant was passed through a 0.22 μm membrane, and the content of each free amino acid fraction was determined by an automatic amino acid analyzer (Hitachi L-8900, Japan).

### 4.6. Exogenous Arginine Treatment

The one-year-old clonal “Shuchazao” was treated with exogenous arginine to test arginine’s effects on the freezing tolerance of tea plants. The 1-year-old clonal tea plants were sprayed with 10 g/L arginine once daily for 4 days, and water was used as the control. To stimulate cold stress conditions, plants sprayed with arginine were left at 4 °C for 7 days and the control was maintained under the same conditions before evaluating freezing tolerance.

### 4.7. Physiological Measurements

After 14 days of cold acclimation (CA), tea seedlings were collected and kept at −4 °C for 12 h (cold stress) and then were kept at 25 °C to recover for 6 h. Tea plants not challenged with cold acclimation (NA) served as controls. After the recovery period, the net photosynthetic rate and maximum photochemical efficiency of photosystem II (Fv/Fm) were measured [61].

Electrolyte leakage was used as a measure of freezing tolerance, and NA-, CA-, and DA- treated tea seedlings were exposed to—4 °C for 12 h. The relative electrolyte leakage rate was determined according to the method described previously [62]. The physiological parameters of the youngest and subsequent second fully expanded tea leaves were measured. The antioxidant activity was determined using peroxidase (POD) assay and a catalase (CAT) assay (Jiancheng, Nanjing, China).

### 4.8. Determination of Polyamines

Free polyamines were extracted and derived as described by Kou et al. with slight modification using 1,6-hexanediamine as an internal standard [53]. In brief, 0.1 g of frozen sample was powdered and extracted with 5% cold perchloric acid containing 500 mg L^−1^ of dithiothreitol and was then derivatized with benzoyl chloride. Subsequently, the derivatized putrescine, spermine, and spermidine were separated and quantified using the Waters AcQuity Arc system (Milford, MA, USA) equipped with a C18 reversed-phase column (4.6 × 150 mm, particle size 3 μm) and a UV detector at 230 nm. The mobile phase was composed of HPLC-grade methanol (eluent A) and water (eluent B) changing from 55%:45% (*v*/*v*, A:B) to 95%:5% in 10 min at a flow rate of 0.7 mL min^−1^.

### 4.9. Gene Expression Analysis

Real-time quantitative PCR analyses were conducted with mRNA from the leaves of six plants from three experimental replicates. A quantitative real-time polymerase chain reaction (qPCR) assay was conducted with CFX96™. The following amplification protocol generated melting curves: 95 °C for 3 min, 40 cycles at 95 °C for 10 s, and 62 °C for 30 s. Each sample was analyzed using three technical replicates. Relative gene expression levels were calculated using the 2^−ΔΔ^ Ct method [63], with GAPDH as the reference gene [64]. The primers for the RT-qPCR are described in Appendix A.

## 5. Conclusions

In this study, the transcriptional and metabolic changes of cold-resistant and cold-sensitive tea plant varieties were analyzed by a comparative metabolomic and transcriptomic analysis during cold acclimation. Our results suggest that the amino acid biosynthesis pathway may dominate the differences in tea plant cold resistance. Arginine is an important metabolite in the tea plant response to CA, which may improve cold resistance by activating the polyamine synthesis pathway and CBF–COR regulation pathway.

## Figures and Tables

**Figure 1 ijms-24-01907-f001:**
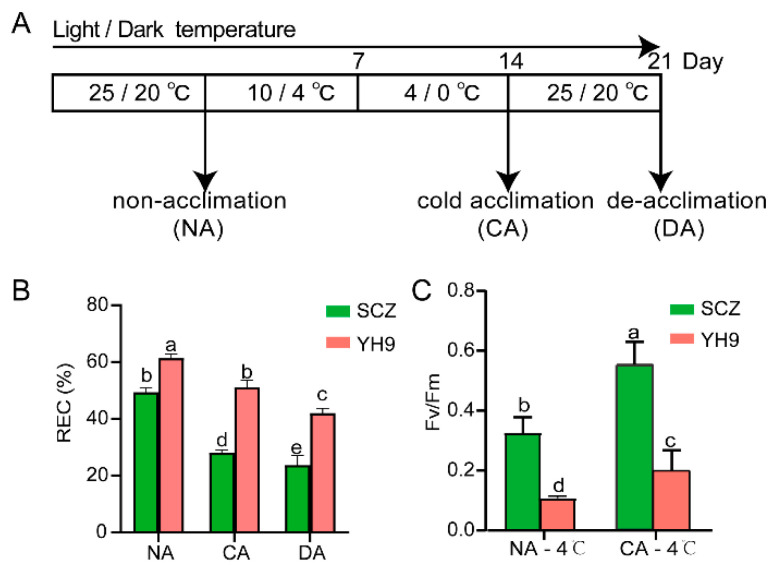
Cold resistance analysis of SCZ and YH9 under cold acclimation. (**A**) Temperature setting and sampling time points during cold acclimation. NA indicate the sampling time points of the control plants; CA, the sampling time points of the respective cold acclimation plants; DA, the sampling time point of de-acclimation plants after a seven-day lag phase at CA. (**B**) The relative electrical conductivity of SCZ and YH9 at NA, CA, and DA stages. (**C**) Statistical analysis of the photochemical efficiency of photosystem II (Fv/Fm) of SCZ and YH9 tissue culture seedlings subjected to freezing (−4 °C, 12 h) at NA and CA stages to assess their cold stress resistance. Error bars indicate the mean ± SD of 16 biological repeats. The different lowercase letters above bars indicate the significance (*p* < 0.05).

**Figure 2 ijms-24-01907-f002:**
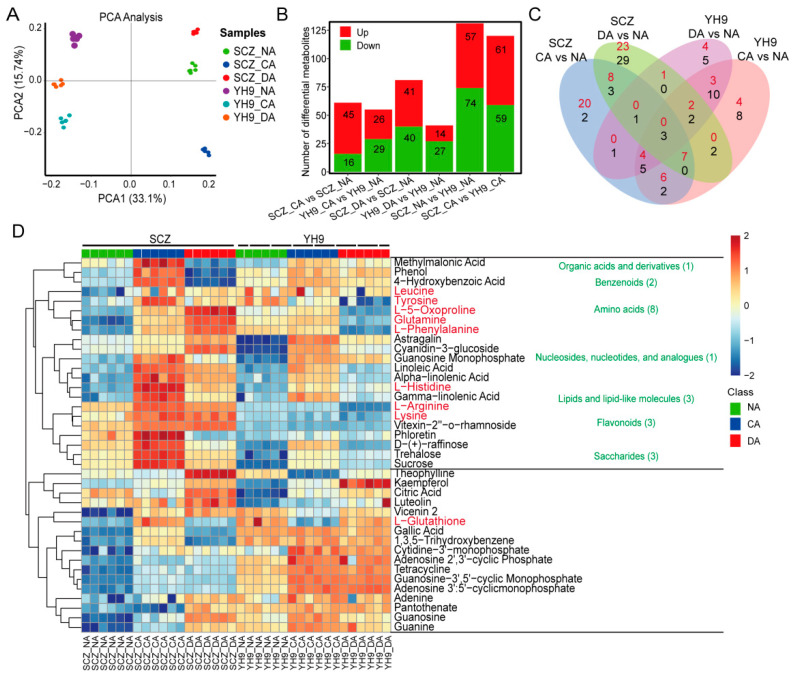
Metabolome analysis of SCZ and YH9 in response to cold acclimation. (**A**) PCA (Principal component analysis) clustering based on metabolome data. (**B**) Number of differentially expressed metabolites in two tea varieties. (**C**) Venn diagrams of differential accumulated metabolites between SCZ and YH9 during cold acclimation (CA and DA). Red represents up-regulation and black represents down-regulation. (**D**) Heatmap of differential accumulated metabolites between SCZ and YH9 during cold acclimation.

**Figure 3 ijms-24-01907-f003:**
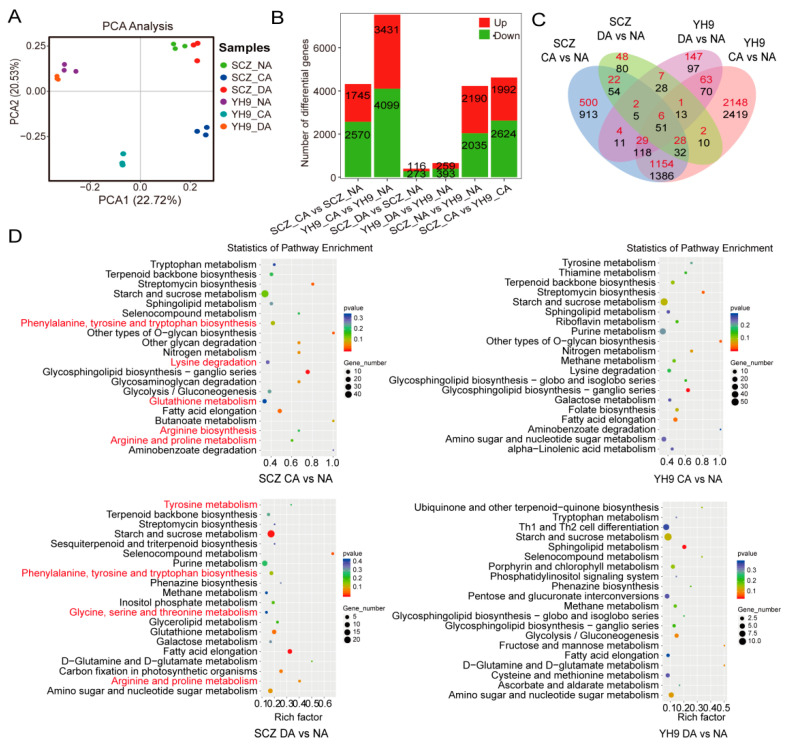
Overview of transcriptomic analysis. (**A**) Principal component analysis of RNA-Seq data of tea plants. (**B**) Number of up-regulated and down-regulated differentially expressed genes (DEGs) in SCZ and YH9. (**C**) Venn diagrams of DEGs between two tea varieties during cold acclimation. Red represents up-regulation and green represents down-regulation. (**D**) KEGG pathway impacted at CA stage in SCZ and YH9.

**Figure 4 ijms-24-01907-f004:**
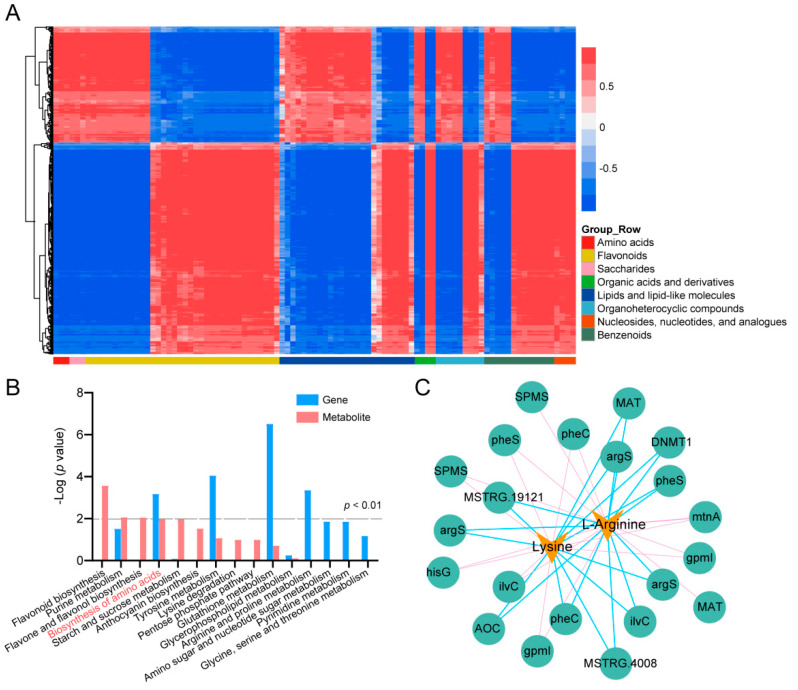
Network analysis of DEGs and differential accumulation metabolites in tea plants at CA stage. (**A**) Heatmap of DEGs and differential accumulation metabolites correlation analysis in the comparison of SCZ_CA and YH9_CA. Horizontal axis represents differential accumulation metabolites; vertical axis represents DEGs. Red represents positive correlation; blue represents negative correlation. (**B**) DEGs and differential accumulation metabolites enrichment in KEGG pathways. (**C**) Correlation network of DEGs and differential accumulation metabolites involved in amino acid biosynthesis pathway.

**Figure 5 ijms-24-01907-f005:**
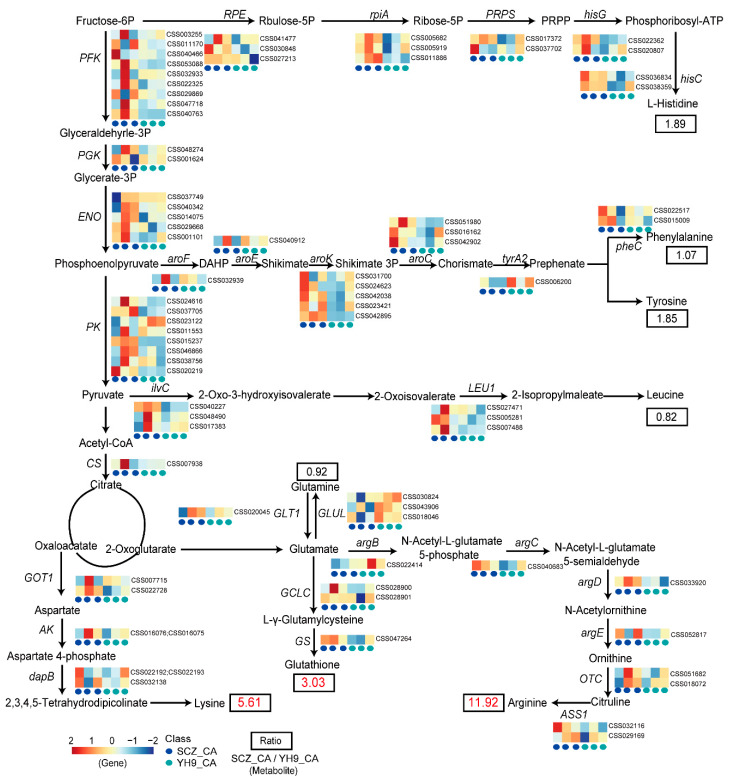
Amino acid biosynthesis pathways in SCZ and YH9 at CA stage. *PFK*, 6−phosphofructokinase 1; *PGK*, phosphoglycerate kinase; *ENO*, enolase; *PK*, pyruvate kinase; *CS*, citrate synthase; *RPE*, ribulose−phosphate 3−epimerase; *rpiA*, ribose 5−phosphate isomerase; *PRPS*, ribose−phosphate pyrophosphokinase; *hisG*, ATP phosphoribosyltransferase; *aroF*, 3−deoxy−7−phosphoheptulonate synthase; *aroE*, shikimate dehydrogenase; *aroK*, shikimate kinase; *aroC*, chorismate synthase; *tyrA2*, prephenate dehydrogenase; *pheC*, cyclohexadienyl dehydratase; *ilvC*, ketol-acid reductoisomerase; *LEU1*, 3−isopropylmalate dehydratase; *GOT1*, aspartate aminotransferase; *AK*, aspartate kinase; *dapB*, 4−hydroxy-tetrahydrodipicolinate reductase; *GLT1*, glutamate synthase; *GLUL*, glutamine synthetase; *GCLC*, glutamate−cysteine ligase catalytic subunit; *GS*, glutathione synthase; *argB*, acetylglutamate kinase; *argC*, N−acetyl−gamma−glutamyl−phosphate reductase; *argD*, acetylornithine aminotransferase; *argE*, acetylornithine deacetylase; *OTC*, ornithine carbamoyltransferase; *ASS1*, argininosuccinate synthase. The heat map was drawn using the Z−score conversion value of transcriptome data. The color scale shows increasing expression levels from blue to red.

**Figure 6 ijms-24-01907-f006:**
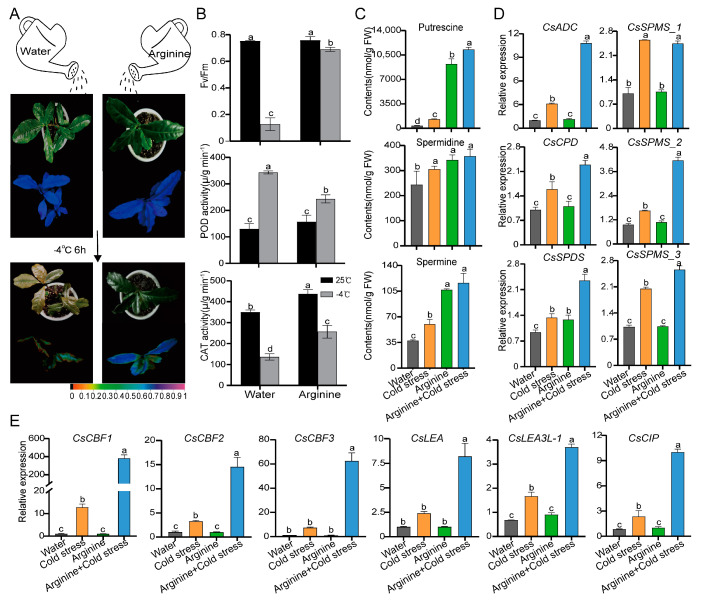
The effect of arginine on cold resistance in tea plants. (**A**) Phenotype of 1% (*w*/*v*) arginine and control group following freezing treatment (−4 °C, 6 h) and Fv/Fm imaging of chlorophyll fluorescence parameters. Purple and blue indicate the normal state of the photosynthetic system, while green and yellow suggest damaged photosystem II. (**B**) Statistical analysis of chlorophyll fluorescence parameter Fv/Fm ratio, POD activity, and CAT activity in tea leaves treated with 1% (*w*/*v*) arginine. (**C**) The effect of arginine on the polyamine levels in tea plant under cold stress. (**D**) Relative expression of genes related to polyamine synthesis in response to arginine application. (**E**) The effects of exogenous arginine on CBFs and COR genes. Different lowercase letters represent a statistically significant difference in the same row (*p* ≤ 0.05).

## Data Availability

Not applicable.

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
