# Peer review of "Integrated Metabolomic and Transcriptomic Analysis Reveals That Amino Acid Biosynthesis May Determine Differences in Cold-Tolerant and Cold-Sensitive Tea Cultivars"

_ijms, 2023, doi:10.3390/ijms24031907_

Round 1

Reviewer 1 Report

Two tea plant varieties were analyzed for cold tolerance, and the SCZ variety was more cold tolerant than the YH9 variety. There is support from the metabolomics and transcriptomics that amino acid biosynthesis is important but their data is descriptive and not particularly conclusive or surprising There seems to be insufficient description of their experiments exogenously applying arginine.

Abstract: “CA” and “DA”, and “CBF-COR”

the amino acid biosynthesis difference may be an indirect effect, and not directly responsible for the cold tolerance. e.g., it could be growth related.

line 57: “are accumulation...”

line 100: it says error bars are three repeats but the image looks like there are 16 repeats?

line 233: “6-phosphofructokinase 1” is an enzyme from central catabolism? how specifically related to amino acid metabolism is this enzyme?

line 327: it cannot be claimed that the amino acid biosynthesis is essential here based on the data...

line 446: more information is needed on the GC-MS methodology used to identify the metabolites? what was the database that was searched against? were any authentic standards used? What is the % probability of the match to the compound in the database? Can the GC-MS distinguish D- and L enantiomers?

Also, the metabolomics does not seem quantitative, and instead, another method is useful to quantify the levels of the amino acids in a broader time course. I think this quantitative analysis of amino acids would give better support to the metabolomics data than RNAseq.

The RNAseq data need to be deposited in a database such as GEO.

From what I can read, the exogenous application of arginine is not described in the M&Ms.

Presumably, the number of functionally characterized genes in tea is small, and then the gene annotations and pathways are predictions based on other related species. This needs to be discussed.

Fig. 1. The Panel “D” needs to come after “C”.

this paper also seems relevant?

doi:10.3390/agriculture10060201

Author Response

Dear Editors and Reviewers:

Thank you for your letter and for the reviewer's comments concerning our manu entitled "Integrated metabolomic and transcriptomic analysis reveals that amino acid biosynthesis may determine differences in cold-tolerant and cold-sensitive tea cultivars". Those comments are all valuable and very helpful for revising and improving our paper, as well as the important guiding significance to our researches. We have studied comments carefully and have made correction which we hope meet with approval. The revised parts are marked in red in the attachment.

Reviewer 2 Report

More detailed information about sample size and data processing is needed.

A control sample (no cold shock treatment) for both varieties should be included to make a fair assessment of the differential gene expression. Variety specific differences already might contribute to a lot of the DEGs or difference in metabolite levels. 

p-value threshold is not a good indicator for RNA-seq based DEG calculation.  FDR based cut-off should be used. 

Would like to know more details about how the PCA plots were generated. 

It is unclear what are the axis for Fig 4A. I am not sure I understand the plot much.

Author Response

(The authors gave the same response as above.)

Round 2

Reviewer 1 Report

I think the authors' revisions are sufficient.

Point 2: The amino acid biosynthesis difference may be an indirect effect and not directly responsible for the cold tolerance. e.g., it could be growth related.

In response to Point 2, the authors replied about a correlation, but this correlation would not necessarily mean causation. It is not clear if the response to Point 2 has been addressed in the manuscript. e.g., has the Tian et al. reference been added?

The tracked changes show that some of the LC-MS methodology has been deleted. The tracked changes should be re-checked.

Also, in other places, try to revise the manuscript in response to the reviewer's comments instead of just replying to the comment.

Author Response

Dear Editors and Reviewers:
Thank you for your letter and for the reviewer's comments concerning our manuscript entitled "Integrated metabolomic and transcriptomic analysis reveals that amino acid biosynthesis may determine differences in cold-tolerant and cold-sensitive tea cultivars". Those comments are all valuable and very helpful for revising and improving our paper, as well as the important guiding significance to our researches. We have studied comments carefully and have made correction which we hope meet with approval. Revised portion are marked in red in the paper. Please see the attachment for specific revision details.
